# Determination of the Highly Sensitive Carboxyl-Graphene Oxide-Based Planar Optical Waveguide Localized Surface Plasmon Resonance Biosensor

**DOI:** 10.3390/nano12132146

**Published:** 2022-06-22

**Authors:** Chien-Hsing Chen, Chang-Yue Chiang

**Affiliations:** 1Department of Biomechatronics Engineering, National Pingtung University of Science and Technology, Pingtung 91201, Taiwan; garychc@mail.npust.edu.tw; 2Graduate School of Engineering Science and Technology and Bachelor Program in Interdisciplinary Studies, National Yunlin University of Science and Technology, Yunlin 64002, Taiwan

**Keywords:** graphene oxide, waveguide, gold nanoparticle, localized surface plasmon resonance, anti-IgG-IgG, biotin-streptavidin, affinity constant

## Abstract

This study develops a highly sensitive and low-cost carboxyl-graphene-oxide-based planar optical waveguide localized surface plasmon resonance biosensor (GO-OW LSPR biosensor), a system based on measuring light intensity changes. The structure of the sensing chip comprises an optical waveguide (OW)-slide glass and microfluidic-poly (methyl methacrylate) (PMMA) substrate, and the OW-slide glass surface-modified gold nanoparticle (AuNP) combined with graphene oxide (GO). As the GO has an abundant carboxyl group (–COOH), the number of capture molecules can be increased. The refractive index sensing system uses silver-coated reflective film to compare the refractive index sensitivity of the GO-OW LSPR biosensor to increase the refractive index sensitivity. The result shows that the signal variation of the system with the silver-coated reflective film is 1.57 times that of the system without the silver-coated reflective film. The refractive index sensitivity is 5.48 RIU^−1^ and the sensor resolution is 2.52 ± 0.23 × 10^−6^ RIU. The biochemical sensing experiment performs immunoglobulin G (IgG) and streptavidin detection. The limits of detection of the sensor for IgG and streptavidin are calculated to be 23.41 ± 1.54 pg/mL and 5.18 ± 0.50 pg/mL, respectively. The coefficient of variation (CV) of the repeatability experiment (sample numbers = 3) is smaller than 10.6%. In addition, the affinity constants of the sensor for anti-IgG/IgG and biotin/streptavidin are estimated to be 1.06 × 10^7^ M^−1^ and 7.30 × 10^9^ M^−1^, respectively. The result shows that the GO-OW LSPR biosensor has good repeatability and very low detection sensitivity. It can be used for detecting low concentrations or small biomolecules in the future.

## 1. Introduction

The refractive index (RI) sensor based on an optical waveguide has been extensively used in chemical or biological detection [1,2,3]. In terms of the operating principle of optical waveguide sensing technology, the light propagates via the optical waveguide substrate. When the physical property of the external environment changes or the analyte is combined with the surface capture molecule, different interfaces between the periphery of the optical waveguide and the analyte result in RI variation. This leads to a change in the light signal or wavelength shift. Quantification, real-time detection, or binding kinetics of the molecular estimation of the target object or analyte can be performed by measuring the subtle optical signal change of RI [4]. Previous studies have published multiple optical waveguide sensors, such as guided-mode resonance [5,6,7], the Mach–Zehnder interferometer [8,9,10], and the fiber grating [11,12,13]. These were developed using different optical waveguide materials. A quick analysis and excellent detection performance were implemented by measuring the change in RI.

With nanotechnology advances, noble metal nanoparticles are extensively used in physical, biochemical, and biological sensing domains, especially the gold nanoparticle (AuNP) [14,15,16]. The AuNP is characterized by particular optical properties, excellent bio-compatibility, and easy modification. It can be combined with a capture molecule (e.g., DNA [16,17], antibody [18,19,20], enzyme, aptamer [21,22,23], or RNA [24,25,26]) by a covalent bond or noncovalent electrostatic interaction. It is extensively used in biological detection and biochemical sensing domains [24,27]. Many LSPR biosensors have been reported thus far, and the findings show that the LSPR biosensor has good detectability [28]. While testing small molecular and extremely low-concentration samples, the rate of change in the resonant wavelength, resonance strength, and detection sensitivity is too small [29,30,31]. Therefore, increasing the sensitivity of the LSPR sensor and testing extremely low-concentration samples are the key points of this study. However, these LSPR biosensor systems typically require complex manufacturing processes, precise optical alignment systems (high-precision angular-resolving rotary stages or xyz three-axis precision optical fine-tuning stages), laser light sources, etc., which result in bulky and expensive sensing systems, limiting the applicability of optical waveguide sensing technology. Therefore, a simple optical waveguide coupled with a gold nanoparticle (AuNP) sensing system should be developed to enhance its applicability, and the sensitivity of the optical waveguide sensor should be improved by an effective method.

Fortunately, increasing the content of the capture molecule in the biosensor is conducive to increasing the sensitivity of biological detection [32,33,34]. Graphene oxide (GO) is a derivate of graphene, which is one-atom thick and has a 2D structure, comprising sp^2^ and sp^3^-hybridized carbon atoms and such functional groups as carboxylic acid (O=C–OH), hydroxy (C–OH), phenol, and epoxyl (C–O–C) [35,36]. As the carboxylic acid is distributed over the surface edge, the phenolic hydroxyl and epoxyl exist in the basal plane. GO has an amphiphile of the hydrophilic edge and hydrophobic basal plane, which have stability and dispersion in an aqueous solution and a polar solvent [37,38]. The GO has a large surface area, stable conjugated π bond, and good biocompatibility surfaces, which is beneficial for the detection of covalently linked peptides (–CO–NH–) in biosensor technology [39,40,41]. Furthermore, the abundance of carboxyl groups (–COOH) in GO increases the number of biological recognition molecules, which significantly increases the immobilization of capture probes, leading to a subsequent increase in sensitivity [42,43]. It has been extensively used in biosensors, for example, the fluorescence is combined with GO for sensitive and selective detection of the target of miRNA let-7a, with the limit of detection (LOD) of 7.8 pM [44]. A surface plasmon resonance (SPR) aptamer sensor based on carboxylated GO was developed to test the biological target hCG protein of Downs syndrome. The result showed that the LOD for the buffer solution sample was 1 pM and that for the serum sample was 1.9 pM [45]. In our previous study [46], GO was used to increase the sensitivity of fiber-optic localized surface plasmon resonance. The result showed that the immunoglobulin G (IgG) molecule detection sensitivity was 0.038 ng/mL. The root cause increase in sensitivity is attributed to the increased adsorption of biomolecules on GO and the optical properties of GO.

This study proposes a novel carboxy-graphene-oxide-based planar optical waveguide localized surface plasmon resonance biosensor (GO-OW LSPR biosensor), which involves a functionalized carboxy-GO-modified binding peptide technology. The developed GO-OW LSPR biosensing system uses low-cost spectrally confined light-emitting diodes (LEDs) as light sources and low-cost photodetectors (PD) as light receivers, combined with self-made integrated circuits (including lock-in technology and amplifying circuits) to convert sample-induced local RI changes at the surface of the optical waveguide into corresponding changes in light intensity. In addition, the biosensing chip utilizes the assembly of the fabricated PMMA plastic and optical waveguide (OW) (glass slide), enabling lower fabrication costs and reduced production time. The structure of the sensing chip comprises an optical waveguide (OW) (slide glass) and microfluidic chip (poly (methyl methacrylate), PMMA substrate). It has been suggested that the excitation of guided modes in total internal reflection can drastically increase light/matter interaction measurements [47]. In order to enable efficient light delivery to the optical waveguide, the silver-coated reflective film has been used for the sensing chip. A silver-coated reflective film increases the number of total internal reflections in the optical waveguide. The slide glass surface-modified gold nanoparticle (AuNP) is combined with graphene oxide (GO). As graphene oxide nanomaterials have abundant carboxyl groups (–COOH) that can provide multiple binding sites, they are beneficial to the surface immobilization of biological recognition molecules (such as antibodies, enzymes, DNA, and cells) and increase the number of capture molecules, thereby improving sensing selectivity and sensitivity. The surface immobilization mechanism of the capture molecules in this study is mainly based on the modification method of forming peptide bonds (–CO–NH–) between the carboxyl groups on the surface of GO and the amino groups of the antibody. Compared to the existing biosensing technology (target: IgG or streptavidin), the GO-OW LSPR biosensor proposed in this study has the characteristics of low cost, real-time, simple detection and excellent sensitivity (about pM levels). It can be used in chemical and biological detection in the future.

## 2. Materials and Methods

### 2.1. Materials and Reagents

Hydrogen tetrachloroaurate (III) (HAuCl_4_·4H_2_O, ≥99.9% trace metals basis), trisodium citrate solution (C_6_H_5_Na_3_O_7_, ≥99%), cystamine dihydrochloride (cystamine), *N*-hydroxy-succinimide (NHS), 1-ethyl-3-(3-dimethyl aminopropyl)-carbodiimine hydrochloride (EDC), (3-mercaptopropyl)-trimethoxysilane (MPTMS, >95%), graphite flakes (99% carbon basis), hydrochloric acid, potassium permanganate (KMnO_4_; ≥99%), glycine, ethanolamine (EA), human serum albumin (HAS), goat anti-mouse IgG antibody, mouse IgG (isoelectric point (pI) = 7), high mobility group box 1 (HMGB1), N-(+)-biotinyl-3-aminopropylammonium trifluoroacetate (Biotin-NH_2_), and streptavidin were bought from Sigma-Aldrich (St. Louis, MO, USA). Hydrogen peroxide (H_2_O_2_) and sulfuric acid (H_2_SO_4_; ≥98%) were bought from Fluka (Buchs, Switzerland). Ethanol and acetone were bought from HyBiocareChem (New York, NY, USA). Sodium citrate was bought from J.T. Baker (Phillipsburg, NJ, USA). All aqueous solutions in this study were prepared using the Milli-Q pure water purification system of U.S. Millipore Ltd. (Burlington, MA, USA) (resistivity = 18.2 MΩ/cm). All biological samples were prepared using phosphate-buffered saline (PBS) having a pH of 7.4.

### 2.2. Sensing Chip Fabrication

Slide glass was bought from AS ONE CO., Ltd. (Osaka, Japan) with a refractive index (*n*) of 1.5195 at a wavelength (*λ*) of 532 nm and thickness of 1.0 mm. Optical adhesive (NOA 85) was bought from Norland Products Inc. (New York, NJ, USA). PTFE Extruded Film Tape was bought from 3M (3M, Taiwan). Silver-coated reflective film (*λ* = 550 nm, light reflectivity >97.0%) was bought from Perm Top Co., Ltd (New Taipei, Taiwan). The CO _2_ laser engraving machine (New Taipei, Taiwan) was used to prepare the sensing chip (PMMA plates, *n* ≈ 1.4934, *λ* = 532 nm). The sensing chip with a size of 50 mm × 24 mm × 1.0 mm was fabricated in two layers and had the components of a lower cover slide glass, UV optical adhesive, and an upper cover that included a PMMA substrate of the microfluidic technique (the sample microchannel was 20 mm × 0.2 mm × 0.2 mm (height) with a volume of 50 μL). The slide glass was cleaned in soapy water, deionized water (DI), methanol solution, DI water, ethanol solution, and DI water in the ultrasonicator. The substrate oscillated in each of these solutions for 15 min. Afterward, the slide glass was cleaned with Piranha solution (30% H_2_O_2_ and 70% strong H_2_SO_4_), fully washed with ultrapure water, and dried in the oven. The slide glass surface was cleaned for the second time using oxygen plasma before AuNP-functionalization. The two sides were shaded with PTFE tape, leaving the sensing area (50 mm × 3 mm) to modify the AuNP. After the AuNP modification, the PTFE tape was removed and the UV adhesive was coated to glue the upper-cover PMMA (UV light irradiated for 15 min in the bonding process to harden the glue). Upon the solidification of UV adhesive, methanol and DI water were injected into the sensing area for cleaning. This was to remove the UV adhesive vapor from the sensing area. Then, a complete microfluidic chip was obtained

### 2.3. Preparation of Gold Nanoparticle-Modified Slide Glass

The synthesis and modification processes of spherical AuNPs were about the same as in a previous study [48]. In the modification process, 2% MPTMS was prepared in toluene and pre-hydrolyzed for 12 h. The glass slides were cleaned in the oxygen plasma cleaning machine for 30 min so that the slide glass surface could carry more -OH functional groups. Afterward, the slide glass was soaked in the pre-hydrolyzed 2% MPTMS/toluene solution for 6 h for the silica functional groups’ hydrolysis-condensation reaction. The self-assembled monolayer of MPTMS was formed on the slide glass surface, and the tail end with the –SH functional group was exposed. The slide glass was first cleaned with 1:1 ethanol/toluene solution, then with ethanol and DI water, and then blow-dried by nitrogen. The slide glass was soaked in the AuNPs solution for 30 min, and the AuNPs were steadily bonded on the slide glass surface by bonding the S–Au covalent bond. Finally, the slide glass was cleaned with DI water and the AuNPs not bonded on the slide glass were removed. After being blown-dried by nitrogen, the modified AuNP slide glass was used for sensing chip packaging. The absorption spectrum measurement and shape and size verification were performed for the synthetic AuNP and modified AuNP using Jasco V-570 UV–Vis–NIR spectroscopy (Tokyo, Japan) and the JEOL JSM-7610FPlus ultra-high-resolution thermal field emission scanning electron microscope (FESEM, Tokyo, Japan).

### 2.4. Functionalization of Gold Nanoparticle Surface

The GO was synthesized using the modified Hummer method, as mentioned above [49]. The chemical covalent bonding method was used for sensing chip surface functionalization in the experimental process, as shown in Figure 1. Gold nanoparticles were immobilized on the surface of the optical waveguide and then modified with cystamine to enrich amine (–NH_2_) groups for further reaction with epoxy groups on GO. Afterward, the EDC/NHS was used to activate the carboxyl of GO. Subsequently, the immobilization of the antibody was accomplished by forming a peptide bond (–CO–NH–) between the carboxyl group of GO and the amino group of the antibody. First, on the AuNP surface of the sensing area, 50 μL of 0.02 M cystamine solution was added to the sensing chip by using a microinjection needle (HPLC Fixed Syringes, SLC-1F-50, ChromTech, Bad Camberg, Germany) for 30 min of modification to make the surface AuNP-cystamine. Afterward, it was cleaned with DI water to remove the unbonded cystamine. The GO was modified on the AuNP-cystamine surface. The 0.2% GO immobilization time was 1 h. The AuNP-cystamine-GO was formed on the surface, and then the unbonded GO was removed by ionized water. A mixed solution of EDC (0.2 M) and NHS (0.05 M) was used to activate the GO surface –COOH functional group for 20 min. Then, 50 μL of anti-IgG (100 μg/mL) or NH_2_-biotin (80 μg/mL) was added for 1.5 h of incubation to form AuNP-cystamine-GO-anti-IgG and AuNP-cystamine-GO-NH_2_-biotin functionalization. Later, the ionized water was used for cleaning to remove the unbonded anti-IgG or NH_2_-biotin. The aqueous solution with 1 M EA of pH 8.5 reacted for 7.5 min to inactivate the unreacted –COO^−^ to prevent nonspecificity. Finally, the PBS solution was injected into the sensing chip. UV/Vis–NIR spectroscopy, FE-SEM, atomic force microscopy (AFM, Dimension Icon XR, Bruker, Billerica, MA, USA), and Raman microscope spectroscopy (Raman, DXR3 Thermo-Fisher Scientific, Waltham, MA, USA) were used for material analyses in the experimental process.

### 2.5. Sensing System

Figure 2a shows a schematic diagram of the optical sensing system of the GO-OW LSPR biosensor and Figure 2b shows the sensing chip. The commercially available low-cost LED (model IF-E93, Industrial Fiber Optic, Inc., Tempe, AZ, USA) was used as the light source, with a peak wavelength of 530 nm. The LED had an internal micro-lens for efficient coupling. The self-made drive circuit generated a 1 kHz square-wave voltage to drive the LED, and the lock-in technology was used to stabilize signals and increase the signal-to-noise ratio (SNR). The sensing chip holder was processed by high-precision computer numerical control (CNC). The silver-coated reflective film was affixed to the four sides and bottom surface of the fixture for the sensing chip holder by using UV adhesive. The light input and output holes were not shaded. This allowed the light to reach the sensing chip, and the silver-coated reflective film could perform multiple total internal reflections to generate an evanescent wave. The evanescent wave could excite the AuNPs to generate LSPR. The other end was the light exit connected to the light detection. The microinjection needle was used for sample replacement in the experiment (a 50 μL sample was enough for detection). In this experiment, a high-stability photodiode (S1336-18BK, Hamamatsu, Tokyo, Japan) was used for the light detection to convert the light into electrical signals. The self-made light-receiving amplifier circuit (PAC, photodiode current/voltage converter, and amplifier) amplified the voltage (i.e., the real-time light intensity) converted by the NI 9234 data acquisition system into digital signals. The generated signal was analyzed by National Instruments LabVIEW 2019 software (Austin, TX, USA) on the computer.

### 2.6. Sample Preparation

To prepare RI samples, 6.8% to 41.7% sucrose was dissolved in DI water to obtain the aqueous solutions of 1.343, 1.353, 1.363, 1.373, and 1.383 RI. The RI was measured and confirmed by a refractometer. The stock standard solution (100 g/mL) was diluted with PBS and stored in a freezer at −20 °C. The samples were prepared and used in the same week to avoid inactivated IgG or streptavidin inducing inspection errors. The prepared IgG or streptavidin standard solution had a 1 × 10^−^^10^ to 1 × 10^−^^6^ g/mL concentration range. It was stored at 4 °C for further usage. Before quantitative testing the analyte, the PBS buffer of pH 7.4 was injected into the sensing chip as a baseline. A low-concentration to high-concentration detection was performed to obtain the corresponding signal responses. The test was repeated three times for each data point, and the data were represented by an average value and standard deviation (mean ± SD). Finally, Origin 2020b (OriginLab, Northampton, MA, USA) was used for statistical analysis. A linear relationship between signal response and concentration was drawn to obtain the calibration curve.

### 2.7. Regeneration Test

An amount of 10 mM glycine buffer was prepared as the reagent for regeneration experiments. An amount of 75.07 mg of glycine was dissolved in 100 mL of DI water, and the diluted HCl adjusted the solution’s pH to 1.8. The anti-IgG was used as an immobilized recognition molecule on the sensing chip. Then, lgG at a concentration of 1 × 10^−^^7^ g/mL was used for the reaction. After reaching dynamic equilibrium, the glycine buffer of pH 1.8 (10 mM) was injected to damage the noncovalent bond between anti-IgG and IgG. This made the IgG depart from the anti-IgG surface. When equilibrium was reached, the PBS buffer was used for rebalancing and the IgG at the same concentration was injected to perform regeneration experiments five times.

## 3. Results

### 3.1. GO-OW LSPR Biosensor Analysis

The principle of the GO-OW LSPR biosensor suggests that when the light is transmitted inside the optical waveguide substrate, multiple total internal reflections generate evanescent wave energy. The AuNP modified to the optical waveguide substrate surface absorbs the evanescent wave energy and the LSPR effect is excited. The light transmitted in the optical waveguide substrate interacts with the AuNP and decays. On the surface of AuNPs modified on the slide glass surface of sensing chips, as the RI of the external environment increases, the evanescent field on the surface of AuNPs is absorbed and the transmission intensity of OW long-range measurement is reduced. In other words, when the plasma absorbance of AuNPs on the OW increases, the local RI of the AuNP surface also increases. The change in the light intensity is recorded to discuss the interaction of the change in the external environment or biological recognition molecule (e.g., capture antibody, DNA, RNA, primer, or aptamer) and analyte. In this study, the evanescent wave absorption measurement of the OW sensor was quantitatively analyzed. The light signal (I_S_) collected by the sensor immersed in the analyte solution was compared with the sensor’s intensity (I_R_) immersed in a blank solution. The sensor response is defined as (I_R_ − I_S_)/I_R_ = ΔI/I_R_.

The optical waveguide (OW) comprised a slide glass and microfluidic chip (PMMA). The RI of the slide glass was *n* ≈ 1.5195, *λ* = 532 nm, and the RI of PMMA was *n* ≈ 1.4934, *λ* = 532 nm. The numerical aperture (NA), which is the sinusoidal quantity (sin *θ*_0_) of maximum acceptance angle *θ*_0_ = 16.2°, was 0.28. Considering the axial incidence when the maximum acceptance angle *θ_0_* = 16.2°, the frequency of total internal reflection per unit length was about 2 times/cm. When the silver-coated reflective film was affixed to the sensing chip fixture bottom, the frequency of total internal reflection per unit length was about 3 times/cm. According to the calculation results, the light signal intensity could be increased by about 1.5 times.

### 3.2. Material Analysis

UV/Vis-NIR spectroscopy (Jasco V-570, Tokyo, Japan), FE-SEM (JEOL JSM-7610FPlus, Tokyo, Japan), AFM (Dimension Icon XR, Bruker, USA), and Raman spectroscopy (DXR3 Thermo-Fisher Scientific, Waltham, MA, USA) were used for material analysis and verification. Figure 3a shows the absorption spectrum of the synthetic AuNP solution in the UV-Vis range of the spectrum. The absorption peak demonstrated a maximum at 525 ± 0.6 nm. As for the AuNPs modified to the slide glass surface, the AuNP absorption peak occurred at 545 ± 0.8 nm. Figure 3b shows an SEM image of AuNPs on the sensing chip surface observed through FESEM. It was observed that the nanoparticles were spherical and not aggregated. The AuNPs in the FESEM image were used for particle size analysis. The result shows that the mean particle size was 18.09 ± 2.35 nm. Figure 3c shows the FESEM image of the GO. The result shows a typical thin, corrugated, and papery GO structure [46]. Figure 3d shows an FESEM image of the modified AuNPs-GO on the sensing chip surface observed through FESEM. It was observed that the surface of AuNPs was covered with a thin layer of GO, forming the coexistence of AuNP/GO film. Figure 3e shows the surface roughness of the sensing chip measured by AFM, and how the GO is distributed over the sensing area was visualized. Figure 3e shows that the surface roughness of AuNPs modified in the sensing area was 17.6 nm. Figure 3f shows that the surface roughness of the representative area of AuNP/GO modified in the sensing area was 27.1 nm. The surface roughness was increased by 9.5 nm, and the scanning surface of all images was 10 μm × 10 μm. These values and AFM images verified the existence of AuNP/GO. Figure 3g shows that the Raman spectrum analysis was performed to study the structural features of GO and AuNP-cystamine-GO sheets. The Raman spectrum of GO film showed two main peaks at ~1360 cm^−1^ (D band) and ~1580 cm^−1^ (G band). Among them, D was derived from the vibration of end carbon in the plane with dangling bonds of disordered graphite, and G was derived from in-plane stretching of the ordered sp^2^-bonded carbon atom in a hexagonal lattice. It was observed in the modified AuNP-cystamine-GO film that the redshift in the Raman spectrum was enhanced, the Intensity of D band I (I_D_)/Intensity of G band I (I_G_) ratio was increased by about 1.07, and the original GO sheet was about 0.945. The AuNP-cystamine-GO nanocomposite was successfully obtained by using the said characterization method.

### 3.3. Optical Waveguide Localized Surface Plasmon Resonance RI Sensor

The previous report indicated that the LSPR absorption peak of AuNPs has a redshift and rise in absorbance as the dielectric constant of the external environment increases [14,50,51]. In addition, according to Mie/Drude theory, the LSPR absorption peak of AuNPs shifts toward a long wavelength as the dielectric constant of the external environment increases [52]. To probe the sensing capability of AuNPs for the RI change in the external environment, sucrose solutions of different refractive indexes were prepared (1.343 RIU to 1.383 RIU). First, the sensitivities of the UV-Vis spectroscopy and the self-mounted OW RI sensor for the RI in the ambient environment were tested and compared. Sucrose solutions of 1.333 RIU to 1.383 RIU were used for sensor resolution detection. When UV-Vis spectroscopy was used with solutions having different refractive indices, the absorbance increased. Overall continuous detection results were stacked up (Figure 4a). Finally, after fixing the observation wavelength at 530 nm to perform the linear regression of the RI with absorbance, a linear relationship was obtained (R = 0.9923) with a sensor resolution of 0.031 RIU^−1^ (Figure 4b).

The GO-OW LSPR sensor developed by our team was used to test the RI of the ambient environment. The optical receiver used in this study was a photodiode. It can receive the emitted light intensity from the OW sensor and export the value as a potential signal. A time-varying potential value was obtained by data acquisition software. The signal jitter of the sensing system can be observed in the captured value, i.e., the system stability. In addition, the sensor resolutions before and after usage of the silver-coated reflective film were measured and compared. The signal response was recorded to verify the sensor resolution. As shown in Figure 5a,b, when the RI of the external environment increased, the sensing system was immediately coupled with light signal attenuation. It was observed that the overall signal reductions (ΔI) before and after usage of the silver-coated reflective film were 0.058 and 0.091, respectively. This result shows that the signal variation after using the silver-coated reflective film was 1.57 times that before using the silver-coated reflective film, matching the calculation result. Therefore, the silver-coated reflective film was used as the subsequent experimental condition. The environmental RI was drawn using a balanced signal in Figure 5a,b, and the result in Figure 5c was obtained. The result shows that the light signals before and after usage of the silver-coated reflective film had a good linear relationship to RI, with the R values of 0.9994 and 0.9997, respectively. The noise of the system with the silver-coated reflective film was 4.6 × 10^−6^ V. The detection resolution was 2.52 ± 0.23 × 10^−^^6^ RIU (sensor resolution = 3 × 4.6 × 10^−^^6^/5.477), and the sensitivity of the relative signal change to the environmental RI was 5.477 RIU^−^^1^. This means that whenever the RI was changed by 0.1 RIU, the light signal was attenuated by 54.7%. The system’s noise without the silver-coated reflective film was 1.2 × 10^−^^5^ V and the detection resolution was 6.88 ± 0.47 × 10 ^−^^6^ RIU. Finally, the sensor resolutions of the developed sensor and the RI sensor reported in references were analyzed. These included the bifacial grating waveguide coupling biosensor (sensor resolution = 4.09 × 10^−^^5^ RIU) [6], surface plasmon resonance sensor (sensor resolution = 7.8 × 10^−^^5^ RIU) [53], prism coupling multimode planar waveguide sensor (sensor resolution = 3 × 10^−^^5^ RIU) [54], fiber optic particle plasmon resonance sensor (sensor resolution = 6.23 × 10^−6^ RIU) [19], and tilted conical fiber sensor (sensor resolution = 8.7 × 10^−^^6^ RIU) [55]. The above references show that the sensor resolution of the developed sensor was the same order of magnitude. Although this result shows that the sensor sensitivities before and after usage of the silver-coated reflective film had very little difference, unique advantages in the enabled efficient light [56] and dip biosensor [57,58] could be realized by the metallic coating on the surface of the optical waveguide. These results proved that the implemented RI sensitivity and linear range are applicable to chemical analyses and biomedical detections.

### 3.4. Nonspecific Adsorption and Specificity Tests

To validate the sensing result, the nonspecific adsorption and specificity tests were performed before IgG detection. First, the AuNPs were modified to the OW substrate. The cystamine and GO were modified by chemical bonding gradually. The surface of AuNPs was accompanied by the –COOH group, and the –COOH group was activated by EDC/NHS. Afterward, the anti-IgG was incubated on the sensing chip surface, and the unreacted –COO^−^ group was encased and intercepted by diethanolamine. The prior study [19] indicated that nonspecific adsorption could be prevented, and the AuNP-cystamine-GO-anti-IgG chip could be completed, as shown in Figure 6. Afterward, the nonspecific adsorption and specificity tests were performed to validate the sensing result. As shown in Figure 7, there was no change in the signals measured by HSA and HMGB1. This was the same as the baseline PBS, which means there was no nonspecific adsorption and specificity bonding. The signal change measured by the injected IgG was induced by the specific bonding event with the capture molecule (anti-IgG).

### 3.5. GO-OW LSPR Sensor of Bioaffinity Interactions

The anti-IgG-IgG or biotin-streptavidin was used for real-time specific capture molecule−analyte interactions to prove the detection performance and sensitivity for biological target objects. In the experimental process, PBS buffer solution was injected into the microflow chip until a stable baseline (I_0_) result was observed (measurement time was 600 secs; signal stability was 0.0064%). From low- to high-concentration-analyte (IgG or streptavidin) detection (1 × 10^−^^10^ to 1 × 10^−^^6^ g/mL), the fluid lead each time remained in static mode until the equilibrium was reached in 10 min (as shown in Figure 8a,b). According to the balanced response signal, the light intensity decreased as the analyte concentration increased, and the binding kinetics of the molecular curve was displayed. This phenomenon resulted when the analyte was combined with the capture molecule. The dielectric constant of the AuNP surface and the LSPR absorption band increased. When the signal was smooth, the binding reaction was balanced. Then, the baseline signal (I_0_) was normalized by the analyte balanced signal (I_s_) of different concentrations to obtain the relative signal I_s_ /I_0_. The standard calibration chart was obtained from the relationship between the relative signal (ΔI/I_0_ = I_0_−I_s_/I_0_) and concentration logarithm. Figure 8c,d shows a good linear relationship (correlation coefficient (R), R = 0.9992 for IgG, and R = 0.9932 for streptavidin). The sensitivity was estimated according to the standard calibration chart. The definition of LOD is that a real-time optical response can be determined (intensity monitoring), and when the S/N is 3, the minimum concentration of analyte can be measured. The limit of detections of the GO-OW LSPR sensor for IgG and streptavidin were calculated to be 23.41 ± 1.54 pg/mL and 5.18 ± 0.503 pg/mL, respectively. It was obvious that the GO-OW LSPR biosensor had excellent detection sensitivity. The order of magnitude is the same as the sensitivity of the AuNP-GO-Anti-IgG FOPPR sensor reported earlier [46].

To further estimate the kinetic constant of the GO-OW LSPR biosensor, the method developed by Chang et al. in 2013 was used to estimate the binding kinetics of molecules in order to determine the antigen–antibody affinity constant and binding kinetic constant [59]. The association rate constant (*ka*) and dissociation rate constant (*kd*) derived from linear fitting were 2.78 ± 0.2 × 10^5^ M^−1^s^−1^ and 2.71 ± 0.8 × 10^−2^ s^−1^, respectively. Afterward, the affinity constant *Kf* was calculated by using *ka* and *kd* values (*Kf* = *ka*/*kd*), which was 1.06 ± 0.83 × 10^7^ M^−1^ (sample numbers = 3). The estimated rate constants matched with the rate constants estimated by using the Quartz Crystal Microbalance (*ka* = 3.0 × 10^−5^ M^−1^s^−1^, *kd* = 1.0 × 10^−4^ s^−1^, and *Kf* = 4.5 × 10^7^ M^−1^) [60] and surface plasmon resonance (SPR) sensor (*ka* = 7.14 × 10^5^ M^−1^s^−1^, *kd* = 4.87 × 10^−3^ s^−1^, and *Kf* = 9.65 × 10^7^ M^−1^) [61] in references. Similarly, the *ka* and *kd* of biotin binding with streptavidin were estimated to be 2.65 × 10^8^ M^− 1^s^−1^ and 3.63 × 10^−2^ s^−1^, respectively, while the *Kf* was calculated to be 7.30 ± 0.23 × 10^9^ M^−1^ (sample numbers = 3). The result shows that the sensing platform’s antigen–antibody affinity and binding kinetic constant were close to the measured values of the quartz crystal microbalance (QCM) and SPR. A fall in the range of reference values proves that our biosensor can be used for studying real-time biomolecular interactions:

### 3.6. Reproducibility and Regeneration

The reproducibility and regeneration of the sensing chip were experimentally evaluated in this study. According to the calibration curve in Figure 8c, the IgG of six concentrations was performed three times and the result of CV calculated by the relative signal response was 2.86% to 10.6%. This shows that the sensor had good reproducibility. In the regeneration experiment, the anti-IgG was used as an immobilized capture molecule on a sensing chip. The IgG at a concentration of 1 × 10^−7^ g/mL was used for the reaction. After dynamic equilibrium was reached, the glycine buffer of pH 1.8 (10 mM) was injected to damage the noncovalent bond acting force between anti-IgG and IgG. When the balance was reached, the PBS buffer was used for rebalancing, and the IgG at the same concentration was injected again. The adsorption and desorption were tested five times. According to the real-time signal response in Figure 9a, after glycine buffer desorption and rebalancing, the new balanced signal could reach the position of the prior balanced signal intensity, and the anti-IgG and IgG had the same signal reduction. As shown in Figure 9b, the CV of five groups of balanced signals (IgG (eq)) of adsorption was 0.029%, and the CV of the balanced signal (glycine buffer (eq)) of desorption tests was 0.047%. This means that the sensing chip had good regeneration.

## 4. Conclusions

A GO-OW LSPR biosensor was successfully developed in this study, which is a system based on the measurement of light intensity change. It was used for quantitative testing of the RI, and the detection resolution was 2.52 ± 0.23 × 10^−6^ RIU. In the biochemical sensing system, the limits of detections of the GO-OW LSPR sensor for IgG and streptavidin were calculated to be 23.41 ± 1.54 pg/mL and 5.18 ± 0.503 pg/mL, respectively. The CV of the reproducibility experiment was lower than 10.6%. The results illustrate that the GO modified in the AuNP-GO nanocomposite showed the highest biological affinity. The GO-OW LSPR biosensor had good sensitivity, low detection, and quick response. The developed biosensor will be helpful in the low-concentration detection of specific proteins or the detection of small biomolecules in the domains of environment, food, and biomedicine.

## Figures and Tables

**Figure 1 nanomaterials-12-02146-f001:**
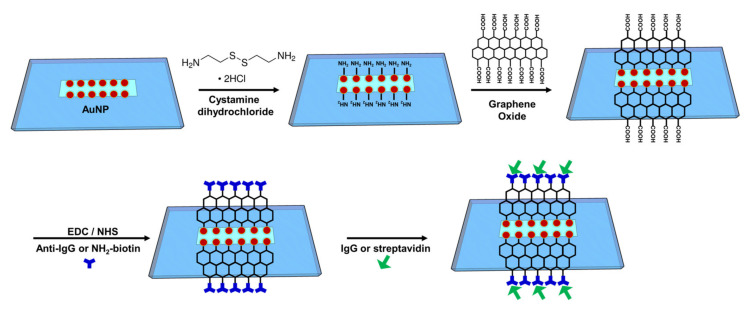
Schematic representation of the AuNP-cystamine-GO-anti-IgG and AuNP-cystamine-GO-NH_2_-biotin functionalization setup in the GO-OW sensor chip. EDC: 1-ethyl-3-(3-dimethylaminopropyl) carbodiimide hydrochloride; NHS: *N*-hydroxy-succinimide.

**Figure 2 nanomaterials-12-02146-f002:**
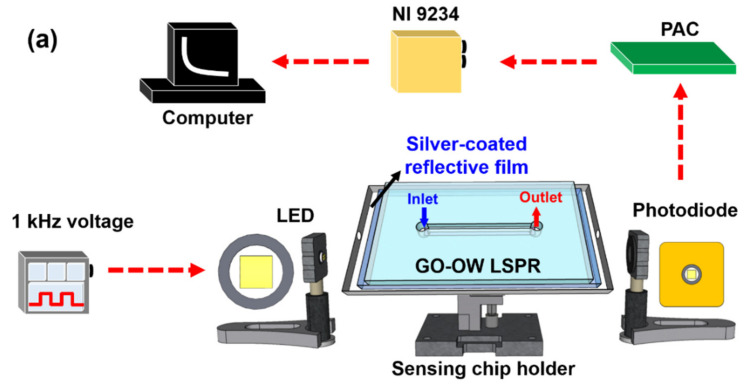
GO-OW LSPR sensor. (**a**) Schematic representation of the experimental setup of the sensing system, and (**b**) sensing chip.

**Figure 3 nanomaterials-12-02146-f003:**
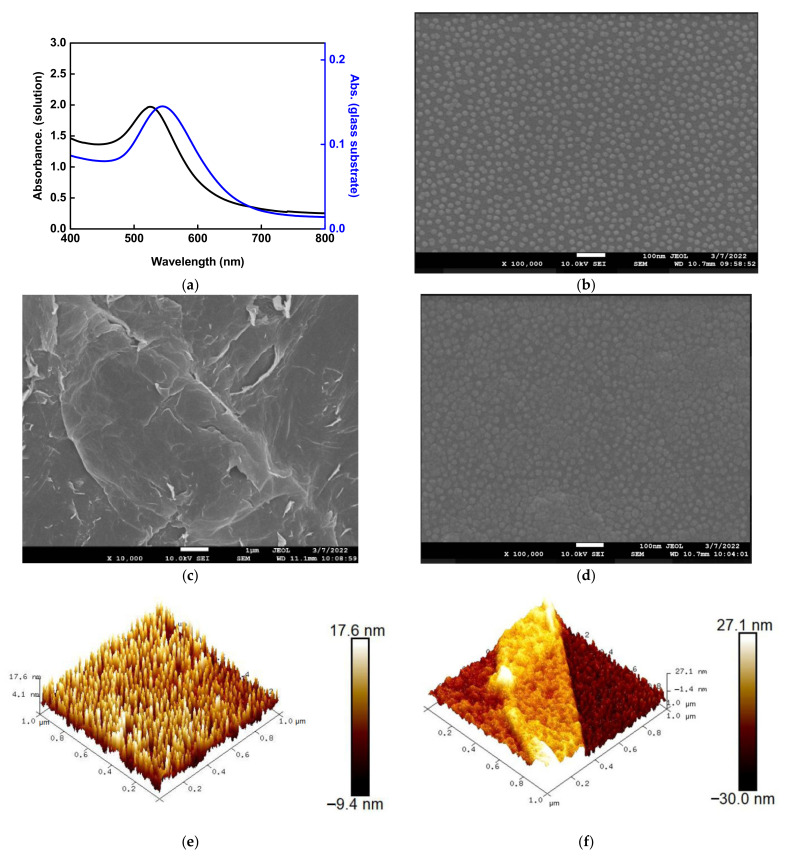
Structural characterizations of materials. (**a**) Absorption spectra of Au nanoparticles (AuNPs) in aqueous medium and the AuNP modified to the slide glass surface measured by UV-Vis spectrum in the visible region; (**b**) SEM image of AuNPs on the sensing chip surface; (**c**) SEM image of graphene oxide (GO); (**d**) SEM image of AuNPs/GO on the sensing chip surface; (**e**) AFM image of AuNPs surface roughness; (**f**) AFM image of AuNP/GO surface roughness; (**g**) Raman spectra of GO and AuNP-GO.

**Figure 4 nanomaterials-12-02146-f004:**
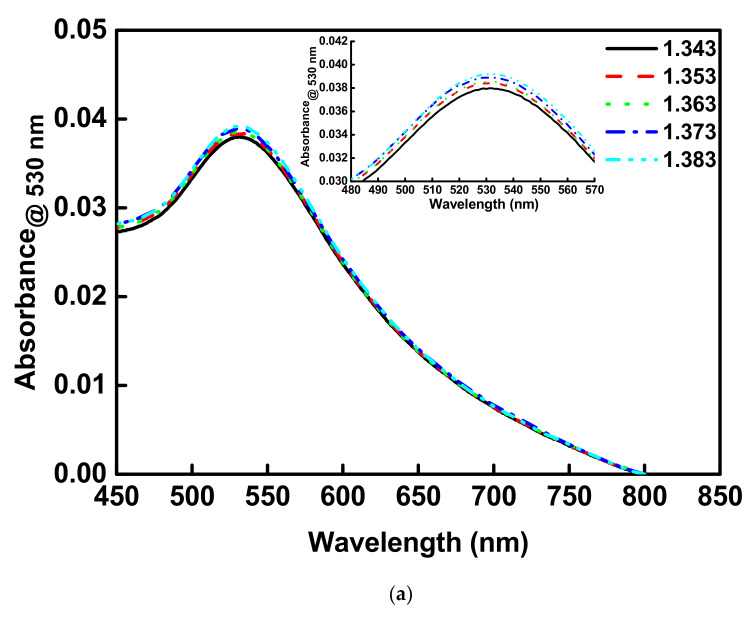
Optical properties of AuNPs on slide glass surface. (**a**) The RI samples at different concentrations were used for UV/Vis-NIR spectroscopy measurement. (**b**) Calibration curves of absorbance vs. refractive index for the UV/Vis-NIR spectroscopy.

**Figure 5 nanomaterials-12-02146-f005:**
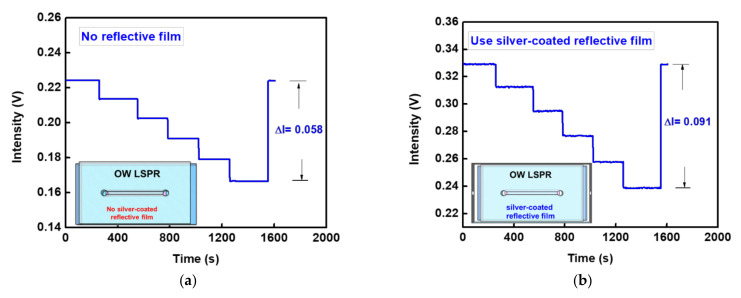
Real-time responses of OW LSPR sensing system for solutions with different RIs showing variation in intensity. (**a**) Different RIs are detected without reflective film (insets in panels: no reflective film added to the sensing chip). (**b**) Different RIs were detected using a silver-coated reflective film (insets in panels: add silver reflective film to the sensing chip). (**c**) Calibration curves of sensor response vs. refractive index before and after using the silver-coated reflective film (sample numbers = 3).

**Figure 6 nanomaterials-12-02146-f006:**
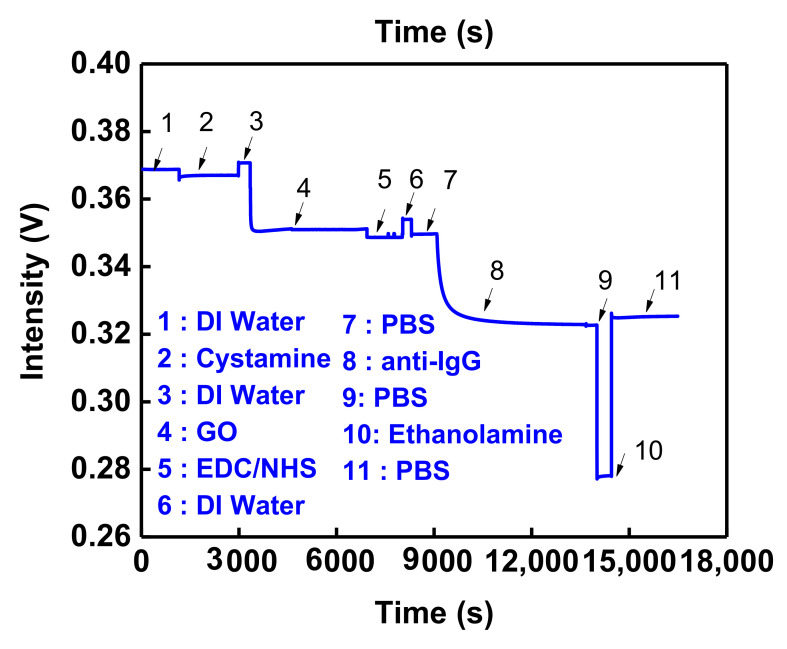
Real-time functionalization of AuNP-Cys-GO-anti-IgG chip.

**Figure 7 nanomaterials-12-02146-f007:**
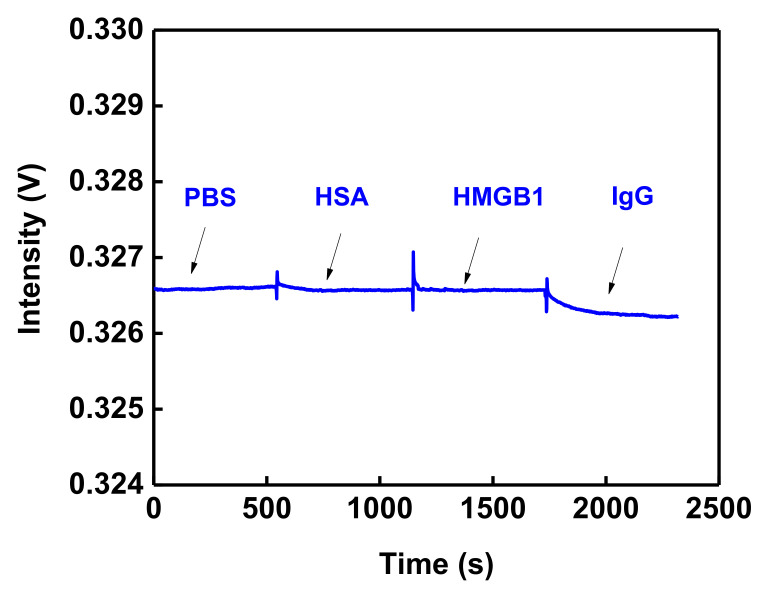
Nonspecific adsorption and specificity tests. The sensorgram of IgG antibody-functionalized GO-OW sensor in response to HSA (1.0 × 10^−6^ g/mL), HMGB1 (1.0 × 10^−6^ g/mL), and IgG (1.0 × 10^−10^ g/mL) solutions.

**Figure 8 nanomaterials-12-02146-f008:**
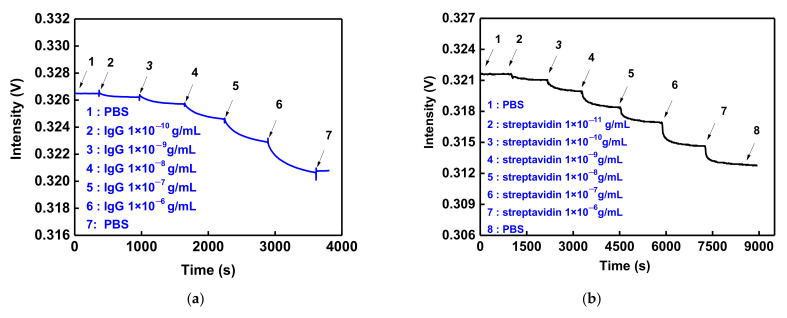
The real-time detection and sensitivity of GO-OW LSPR sensor. (**a**) Temporal response of the anti-IgG-functionalized GO-OW sensing chip signal with serial injection of standard solutions with different IgG concentrations of (1) 1.0 × 10^−10^, (2) 1.0 × 10^−9^, (3) 1.0 × 10^−8^, (4) 1.0 × 10^−7^, and (5) 1.0 × 10^−6^ g/mL. (**b**) Temporal response of the biotin-functionalized GO-OW sensing chip signal with serial injection of standard solutions with different streptavidin concentrations of (1) 1.0 × 10^−11^, (2) 1.0 × 10^−10^, (3) 1.0 × 10^−9^, (4) 1.0 × 10^−8^, (5) 1.0 × 10^−7^_,_ and (5) 1.0 × 10^−6^ g/mL. (**c**) Calibration curve for IgG by anti-IgG-functionalized GO-OW sensing chip. (**d**) Calibration curve for streptavidin by biotin-functionalized GO-OW sensing chip.

**Figure 9 nanomaterials-12-02146-f009:**
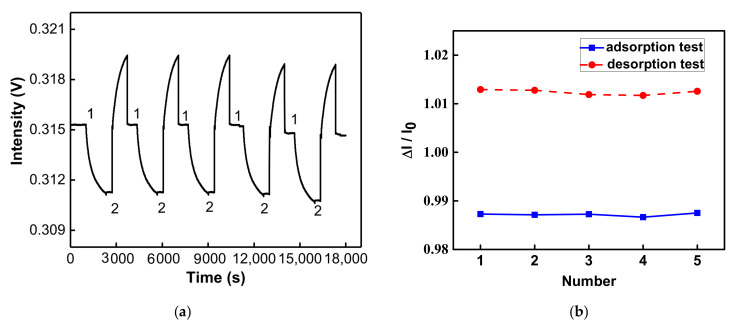
Regeneration study of anti-IgG-functionalized GO-OW sensor chips. (**a**) Adsorption and desorption real-time signal response (No. (1) is to inject the IgG; No. (2) is to inject the glycine buffer). (**b**) Relative signal response values for adsorption and desorption tests (sample numbers = 5).

## Data Availability

Not applicable.

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
