# Peer review of "Determination of the Highly Sensitive Carboxyl-Graphene Oxide-Based Planar Optical Waveguide Localized Surface Plasmon Resonance Biosensor"

_nanomaterials, 2022, doi:10.3390/nano12132146_

Round 1

Reviewer 1 Report

This manuscript describes the fabrication and characterisation of a GO-OW LSPR biosensor which is

based on the measurement of light intensity change. The result shows that the signal variation of the system with the silver-coated reflective film is 1.57 times that of the system without silver-coated reflective film. It was used for quantitative testing  of RI, and the detection resolution is 2.52 ± 0.23 × 10-6 RIU. The limit of detections of the GO-OW LSPR sensor for IgG and streptavidin were calculated to be 23.41 ± 1.54 pg/mL and 5.18 ± 0.503 pg/mL, respectively. The CV of the reproducibility experiment was 2.86 % to 10.6%. There are two major issues that need to be properly addressed/clarified as follows.

1.       The work currently fails to demonstrate “Compared  to the existing IgG sensing technology, the GO-OW LSPR biosensor proposed in this study  has the characteristics of low cost, real-time, simple detection, and excellent sensitivity”. There is no evident for the conclusion of lower cost compared to the existing system. The literature also showed real-time and simple detection; how better is this work? The main point is the sensitivity: The signal variation of the system with the silver-coated reflective film is 1.57 times that of the system without silver-coated reflective film. However, the relative change of intensity showed very little difference (Figure 5c) due to the variation of Io. This leads to the conclusion that the excellent sensitivity is unclear. I suggest to make comprehensive comparison between the performance of device by this work and the literature. This helps readers grasp advances made by this work if any.

2.       No scientific explanation for why is the role of GO nanomaterials. Why did the authors utilize GO-OW and what is the mechanism of improvement?

Author Response

Dear Reviewer:
Thank you for giving us the opportunity to revise our manuscript. We appreciate the careful review and constructive comments from the editors and reviewers. We have taken these comments and suggestions fully into account and revised the manuscript accordingly. Our point-by-point responses to the reviewer comments are appended to this letter, beginning on the next page. List of changes made in the revised manuscript based on reviewer comments. In the following, the reviewer comments are in black Arial font while our responses are in blue Times Roman font. We have also highlighted the changes made in the revised manuscript (with changes in red Times Roman font). Thank you very much for your kind consideration of this paper.

Reviewer 2 Report

This article reports on the development and research of an inexpensive and highly sensitive planar optical waveguide based on carboxyl-graphene oxide with a localized surface plasmon resonance biosensor. The conducted studies show that the proposed biosensor has good sensitivity, low level of detection and fast response. The developed biosensor can be used to detect low concentrations of specific proteins or to detect small biomolecules in the areas of environment, food and biomedicine. The results obtained are new and interesting. However, the article requires minor revision before it is accepted for publication, taking into account the following comments:

1.    The abbreviation IgG was first introduced on page 13 in the caption to fig. 8. Abbreviations must be entered on the page where they are mentioned for the first time, including in the abstract.

2.    All physical designations such as,  n, Rs, ka, kd, Kf , etc, must be in italics. In some cases the authors used a space between the main designation and subscript (see, for example, page 10 – R s,  or page 12 - DI/I 0) . This space must be removed.

3.    The designation n is used by the authors for the refractive index.  Therefore, it is not clear what n means in the caption to Fig. 5 (n=3) and to the caption to Fig. 9 (n=5)?

4.    The first sentence in the caption to Fig. 9 should be moved to the main text of the paper. Instead, a correct description of what is actually shown in fig. 9(a), must be introduced.

5.    In the "Materials Analysis" section, it would be useful to indicate the brand (or manufacturer) of the instruments used for material characterization.

6.    English needs to be checked very carefully, as there are numerous typos and spelling errors. In addition, many sentences are incorrect and difficult to understand, and some sentences are not typical of scientific English. See, for example, the following:

a)      page 3, line 134 – replace “The clean slide glass was cleaned in” with “The glass slides were cleaned in”;

b)      page 7, lines 251-254 – replace these sentences with “The UV/Vis-NIR spectroscopy, FE-SEM, AFM, and Raman spectroscopy were used for material analysis and verification. Figure 3(a) shows the absorption spectrum of synthetic AuNP solution in UV-Vis range of spectrum. The absorption peak demonstrates a maximum at 525 ± 0.6 nm. Figure 3(b) shows a SEM image of AgNPs on the sensing…..”;

c)      Fig.3 caption (and similar cases) – correct spelling in “nanopar-ticles”, “meas-ured” and introduced space after (g), change “Raman spectrum” to “Raman spectra”.

d)      Page 9, line 288 – correct “to-wards”;

e)      Fig. 4(a) caption – replace with proper figure caption;

f)       Page 11, line 340 (and similar cases) – rather than use “AuNPs were fixed to the OW substrate” use  AuNPs were deposited on the OW substrate” or AuNPs were formed on the OW substrate”

Author Response

(The authors gave the same response as above.)

Round 2

Reviewer 1 Report

The authors have attempted to address my comments. Some improvements have been made